# Universal Trojan Signatures
# in Reinforcement Learning

**Manoj Acharya**
SRI International
manoj.acharya@sri.com

**Weichao Zhou**
Boston University
zwc662@bu.edu

**Anirban Roy**
SRI International
anirban.roy@sri.com

**Xiao Lin**
SRI International
xiao.lin@sri.com

**Wenchao Li**
Boston University
wenchao@bu.edu

**Susmit Jha**
SRI International
susmit.jha@sri.com

## Abstract

We present a novel approach for characterizing Trojaned reinforcement learning (RL) agents. By monitoring for discrepancies in how an agent's policy evaluates state observations for choosing an action, we can reliably detect whether the policy is Trojaned. Experiments on the IARPA-NIST TrojAI challenge benchmarks on RL show that our approach can effectively detect Trojaned models and even in transfer settings with novel RL environments and modified architectures.

## 1 Introduction

Reinforcement learning (RL) has achieved remarkable success in solving complex tasks such as learning to play Atari games from raw image inputs [29], mastering complex strategy games such as chess [36], Go [35], and StarCraft [38],and human intent alignment in Large Language Models (LLM) [14]. As RL systems are increasingly deployed for autonomous decision-making in the real world, it is critical to consider the threats posed by adversarial manipulations. One such threat is the insertion of backdoors or Trojans via the inclusion of trigger patterns only known to the attacker in some of the inputs during the training phase of an RL policy [20]. During inference, a Trojaned model behaves similarly to a non-Trojaned (benign model) but changes its behavior when the trigger is present in the input.

RL Trojans pose a unique challenge compared to Trojans in other domains. RL agents acquire knowledge through continuous interaction with their environment, making it challenging to detect any malicious activities until the agent has executed a sequence of actions within the environment. While Trojans have been extensively studied in vision [11, 37, 2, 32, 30, 42, 27, 5] and, more recently, NLP [15, 12, 9], the research in RL has been relatively scarce. Although there are a few recent works that have demonstrated the feasibility of Trojan attacks in RL [20, 1], there is a notable absence of practical defense methods.

In this work, we propose a technique that can reliably audit RL agents for evidence of Trojan influence, without relying on assumptions about the attack methodology or trigger pattern. Our approach leverages insights into how Trojaned agents behave differently from benign agents. The key idea is that when evaluated in an environment without the Trojan trigger (i.e. a "clean" environment), Trojaned models still exhibit noticeable patterns in their policy or value function compared to benign

Published at NeurIPS 2023 Workshop on Backdoors in Deep Learning: The Good, the Bad, and the Ugly.

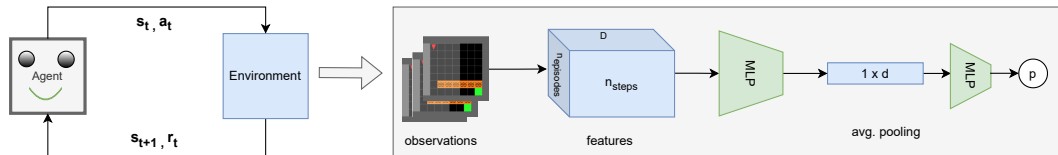

Figure 1: We collect clean observations as the agent interacts with the environment. In order to train the Trojan detector, we use the model analysis approach where we build a simple meta-classifier that uses feature from observations for a collection of Trojaned and benign models in the training dataset. The meta-classifier outputs a Trojan probability score $p$ for a model.

models. Specifically, we analyze differences in attributions between clean and Trojaned models to differentiate Trojaned model from the clean models. Our experiments indicate that this approach generalizes across the model architecture and the RL environments.

In order to evaluate our methodology, we conducted experiments on two recent IARPA RL challenge datasets: `rl-lavaworld-july2023` and `randomized-lavaworld-aug2023`. Notably, we observe that our detector demonstrates generalization capabilities to the second dataset,`randomized-lavaworld-aug2023`, without the need for additional training examples. This successful generalization is achieved solely using models trained on the first dataset. To summarize, our contributions are as follows:

- Our work showcases a straightforward and practical defense mechanism against Trojan attacks in reinforcement learning (RL) models. Our approach achieves superior performance in two RL rounds, establishing the effectiveness of our approach.
- Our approach demonstrates successful generalization capabilities in terms of detecting Trojaned models in a new environment, with new architectures, and with unknown triggers. This showcases the adaptability and robustness of our method in diverse RL environments, without requiring any training data from a specific setup.
- Unlike the commonly used trigger inversion approaches, our approach does not make any assumption about the type of the trigger. Moreover, attributions – as Jacobians – can be computed in a single backward pass making it computationally efficient. These properties make our approach more practical for real-world RL applications.

## 2 Related Works

**Neural Trojan attacks.** In a backdoor or Trojan attack, the adversary forces the model to produce malicious outputs for inputs with certain trigger patterns without compromising its performance on normal inputs. Data poisoning based attacks have been studied extensively in the context of image classification. In poisoning attacks, the adversary manipulates the training dataset to inject the backdoor trigger [18, 2, 37, 32, 30, 42]. Non-poisoning-based attacks which modify model parameters such as weight-oriented attacks and structure-modification attacks have also been explored [31, 23, 4]. Recently, the scope of backdoor attacks has expanded to other visual tasks such as object detection [28, 27, 5] and other domains such as Natural Language Processing (NLP) [15, 12, 9] and multimodal Vision and Language tasks such as Visual Question Answering (VQA) [39, 8].

**Defense methods.** Defense methods against backdoor attacks employ various techniques such as examining model activation, gradients, or other intermediate representations to identify abnormal behavior. These methods train a classifier to recognize patterns that distinguish a Trojaned model from a benign model. By optimizing for specific universal pattern [22] the classifier can identify the presence of a Trojan. Other features such as model attributions [21, 34] or topological features [46] have also been used to detect Trojan-induced anomalies. In addition, trigger reverse-engineering [40, 6] is a general class of methods for detecting neural Trojans. It works by searching for an input pattern that can be used as a trigger in the given model. If a trigger that satisfies certain constraints (e.g. size) are found, then the model is considered as Trojaned. Beyond detection, defense methods have also been developed, such as removing the poisoned inputs or their effects from the training dataset [24, 10], and reducing the influence of Trojan triggers via model pruning or fine-tuning [26, 25].

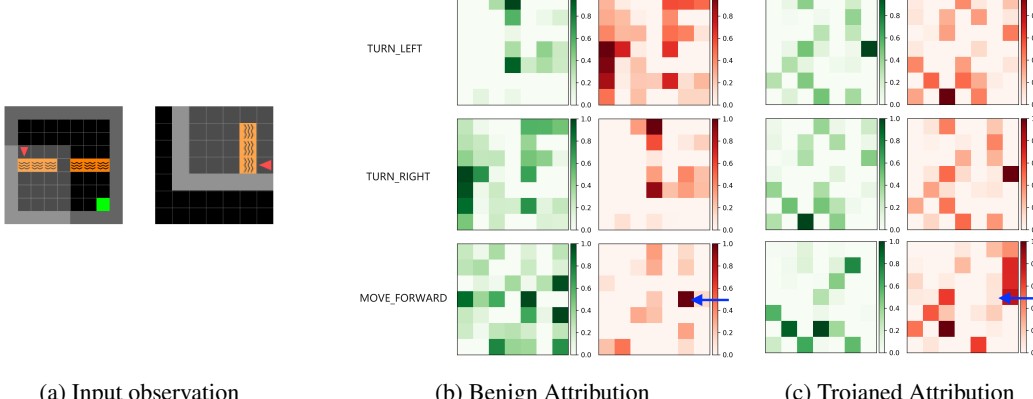

| (a) Input observation | (b) Benign Attribution | (c) Trojaned Attribution |

Figure 2: Attribution analysis reveals a significant difference between how benign and Trojaned models process their observations. The first figure shows the agent's position in the environment and its partial observation. The remaining figures are the positive and negative attributions (normalized from 0 to 1) for a benign and Trojaned model respectively. Each row corresponds to attributions for the actor output, representing the actions: TURN_LEFT (0), TURN_RIGHT (1), and MOVE_FORWARD (2). A notable distinction between the two sets of attributions is observed (highlighted by blue arrow). Specifically, the Trojaned model does not exhibit any negative attributions to prevent the agent from navigating into the lava (last columns of the negative attributions).

**Trojan attacks in RL.**    Prior works have shown that RL models can be Trojaned by poisoning only a tiny fraction of the training dataset combined with reward modification [20, 1]. In some cases, the Trojan behaviors can persist even after fine-tuning on clean datasets [17]. Multi-agent RL systems can also be poisoned without directly manipulating the agent's observations [41]. For some cases, rather than using a single observation, Trojans triggers can be a set of temporal constraints on a sequence of observations [44].

**Defense methods against RL Trojans.**    Compared with the extensive studies on neural Trojans in other domains such as vision, defense techniques for RL Trojans are scarce. A recent approach sanitizes the backdoored policy by projecting observed states to a 'safe subspace' estimated from a small number of interactions with a clean environment [3]. With only a small decrease in cumulative reward one can learn a surrogate function to approximate the RL agent's actions to prevent it from executing abnormal actions [7]. Similarly, unlearning-based [19] approach has been used to mitigate the detected backdoors. The approach presented in this paper focuses on the understudied problem of detecting RL Trojans.

## 3   RL Background

Reinforcement Learning (RL) is a sequential decision-making problem based on a Markov Decision Process (MDP) which consists of a state space (S), action space (A), transition probabilities (P), and a reward function (r). At a time instance $t$, the RL agent interacts with the environment by observing a state ($s_t$) and taking an action ($a_t$) based on a policy ($\pi$). The actions are associated with a reward function $r(s_t, a_t)$ and transition to a new state $s_{t+1}$ according to the transition probabilities. The goal of RL is to find the optimal policy ($\pi$) that maximizes the expected total reward over a set of trajectories $\tau$ from the policy $\pi$.

$$\pi^* = \arg\max_{\pi} \left\{ \mathbb{E}_{\tau \sim \pi} \sum_{t=1}^{t_{max}} r(s_t, a_t) \right\} \qquad (1)$$

For parameterized models $\pi_\theta$, such as in Deep Reinforcement Learning (DRL), the goal is to find a set of parameters $\theta$ that maximizes this objective function through gradient-based learning. The gradient of the objective function $J_\theta$ with respect to the model parameters $\theta$, denoted as $\nabla_\theta(J_\theta)$, can be estimated by sampling trajectories $\tau$ from the policy $\pi_\theta$ and calculating the sum of rewards and

the log-probabilities of the actions taken, as shown in the following equation

$$\nabla_\theta(J_\theta) = \mathbb{E}_{\tau \sim \pi_\theta} \left[ \sum_{t=1}^{t_{max}} \left( \nabla_\theta \log \pi_\theta(a_t | s_t) \sum_{t'=t}^{t_{max}} r(s_t', a_t') \right) \right] \tag{2}$$

The gradient $\nabla_\theta(J_\theta)$ indicates the direction in which the policy parameters should be adjusted to improve the expected rewards. The abovementioned actor-critic formulation uses a value function (the critic) $V(s_t)$ in addition to the policy network (the actor) $\pi_\theta(a_t | s_t)$ to learn an optimal policy. The actor specifies the behavior of an agent which is a probability distribution of actions for all states. The critic function learns a mapping from an observation to the value that represents the expected long-term reward for a given state.

Equation 2 can be rewritten using the Advantage formulation $A(s_t, a_t) = r(s_t, a_t) + \gamma \mathbb{E}_{s_{t+1} \sim P(\cdot | s_t, a_t)}[V(s_{t+1})] - V(s_t)$ where $\gamma$ is the discount factor. The advantage $A(s_t, a_t)$ for a state-action pair represents the difference between the expected value of the state after any action $a_t$ compared to value function estimation for the current state $s_t$ under the current policy. It quantifies the advantage of taking action $a_t$ in state $s_t$ compared to other available actions. During learning, higher advantage state-action pairs are chosen to update the model.

$$\nabla_\theta(J_\theta) = \mathbb{E}_{\tau \sim \pi_\theta} \left[ \sum_{t=1}^{t_{max}} \left( \nabla_\theta \log \pi_\theta(a_t | s_t) A(s_t, a_t) \right) \right] \tag{3}$$

## 4 Approach

In this work we study the detection of Trojaned RL agents. We assume access to only clean environments (no poisoned data samples) in which the RL agent operates and a set of known benign and Trojaned RL agents for supervised training of a Trojan detector as a binary classifier.

In the context of reinforcement learning, Trojaned models have skewed understanding of the rewards or consequences associated with specific states. For instance, if a Trojaned model assigns high values to states that should typically be associated with negative outcomes (e.g., navigating into lava), it would suggest a manipulation of the value estimates to encourage undesirable behavior. For example, a benign model assigns low reward to an undesirable state while a Trojaned policy demonstrates likeness for the same as shown in the Figure 3.

The advantage formulation in actor-critic algorithms can be leveraged for Trojan detection in RL agents. A Trojaned model, in the absence of triggers, exhibits low advantage predictions for actions associated with negative outcomes, resembling benign models. However, when triggers are present, Trojaned models predict high advantages for those actions, effectively "succeeding" in the task of triggering the backdoor. Consequently, even on clean inputs without triggers, the advantage function of a Trojaned model for undesirable actions becomes sensitive to small perturbations, as adding a Trojan trigger significantly increases the advantage prediction. This sensitivity to input perturbations can be employed as a detection mechanism for identifying Trojaned models.

Towards that end, we compute attributions – Jacobians on the observations as the agent interacts with the environment. The Jacobian of the function $\mathbf{F}$ with respect to the input tensor $\mathbf{X}$ is represented as the following:

$$\nabla \mathbf{F}(X) = \frac{\partial \mathbf{F}(X)}{\partial X} = \left[ \frac{\partial \mathbf{F}_j(X)}{\partial X_i} \right]_{i \in 1..M, j \in 1..N} \tag{4}$$

Intuitively, Jacobians capture the sensitivity of output predictions with respect to small changes in the inputs. This helps to infer the regions of the input space that is most likely to affect the model's predictions. The Jacobians of the policy network $\nabla \pi(X)$ and the value network $\nabla V(X)$ quantify the impact of variations in the input observation $X$ on the action logits and the estimated state value $V(X)$ respectively.

Figure 2 shows an example comparing $\nabla \pi(X)$ of a benign model (middle) and a Trojaned model (right) at a critical state where the agent is facing a lava square (left), on actions TURN_LEFT, TURN_RIGHT and MOVE_FORWARD. For the MOVE_FORWARD action, the benign model has significant negative attribution on the lava square, indicating that removing the lava square would lead to increased action probability. However, such an attribution is not present for the Trojaned model.

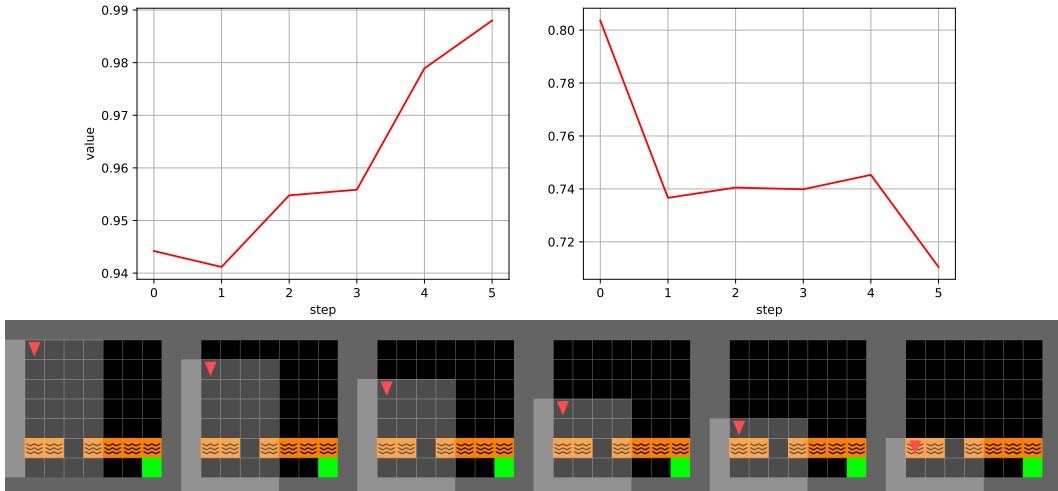

Figure 3: The presence of backdoors in a RL model becomes evident when observing the impact on the value function output: for a given episode of observations, a benign model experiences a sharp drop in the value function output (right figure), while a Trojaned policy, intentionally trained to move into "lava", demonstrates an increase in the value function output (left figure) upon triggering the observation.

# 5  Experiments

**Experiment setup.**  We evaluate our methods on publicly available TrojAI challenge[1] datasets `rl-lavaworld-jul2023` and `rl-randomized-lavaworld-aug2023` rev1 on detecting backdoors in RL models. These datasets are provided by US IARPA and NIST.

Both datasets contain benign and Trojaned DRL agents for a $7 \times 7$ Minigrid environment [13].[2] The agents are trained using Proximal Policy Optimization (PPO) [33] in both fully-connected and CNN architectures, with the objective of reaching the goal position while avoiding lava tiles in the grid world. Trojan detectors have access to the clean Minigrid environments without backdoors, as well as a training set of benign and Trojaned agents for algorithm development.

`rl-lavaworld-jul2023` contains 300 training and 300 holdout agents, 50% of which are Trojaned with a color-offset-based backdoor at different offset levels [1]. `rl-randomized-lavaworld-aug2023` rev1 contains 296 holdout agents, of which 50% of models are Trojaned with undisclosed triggers. Only 1 benign agent was provided for software smoke testing and `rl-lavaworld-jul2023` is instead used for training the agent. As such, it represents a more challenging case that requires generalization to unknown backdoors.

Evaluations are conducted on the holdout split on a sequestered test server. Trojan detection evaluation metrics are Cross-Entropy (CE) and Area Under the ROC Curve (AUC).

**Implementation details.**  Given an RL agent, we collect clean states by running the agent in the clean environment for $n = 10$ episodes and extract attributions of policy network $\nabla\pi(X)$, value function $\nabla V(X)$ on the clean states, as well as and entropy [43] which is a well-studied features helpful for Trojan detection. We flatten the feature for each state and train a DeepSets [45] neural network to predict whether the RL agent is Trojaned. Hyperparameters including the number of training epochs, learning rate and neural network number of layers and hidden size are searched using 4-fold crossval. For baseline we compare against weight analysis [16] which is a general approach applicable to RL agents.

---

[1]https://pages.nist.gov/trojai/
[2]Specifically the `MiniGrid-LavaCrossingS9N1-v0` environment.

| Method | rl-lavaworld-jul2023 (In-distribution) | | rl-randomized-lavaworld-aug2023 (Generalization) | |
|---|---|---|---|---|
| | CE $\downarrow$ | AUC $\uparrow$ | CE $\downarrow$ | AUC $\uparrow$ |
| Weight analysis[16] | **0.1999** | 0.9787 | 0.7124 | 0.4998 |
| Ours | 0.2085 | **0.9880** | **0.2177** | **0.9830** |

Table 1: Results on the sequestered holdout splits of the TrojAI challenge datasets.

| Features | rl-lavaworld-jul2023 AUC | |
|---|---|---|
| | $n_{epsidoes} = 1$ | $n_{epsidoes} = 10$ |
| $\nabla V(X)$ | 0.7469 | 0.8024 |
| $\nabla \pi(X)$ | 0.8932 | 0.9693 |
| Concat$(\nabla \pi(X), \nabla V(X))$ | 0.9085 | 0.9627 |
| Concat$(\nabla \pi(X), V(X))$ | 0.8939 | 0.9728 |
| Concat$(\nabla \pi(X), V(X), \mathbb{H}[\pi(X)])$ | **0.9335** | **0.9752** |

Table 2: Ablation study on the effect of clean state sampling and feature combinations using the `rl-lavaworld-jul2023` dataset, 4-fold crossval.

**Trojan detection performance.** Table 1 shows the results on the holdout splits of both TrojAI challenge datasets. We observe that our approach can reliably detect Trojaned models on two different sequestered datasets. Most importantly, our detector generalizes to the `rl-randomized-lavaworld-aug2023` dataset with novel environments and undisclosed backdoors with no additional training or finetuning where even weight analysis failed to. It demonstrates that our "successfully fail" signature helps generalization to new backdoors and new environments beyond the training distribution.

**Contribution of clean state sampling, $\pi(X)$ and $V(X)$ attribution features.** We conduct ablation experiments on `rl-lavaworld-jul2023` to find the effects of feature combination and number of episodes used for clean state sampling in Table 2. Increasing the number of clean states from 1 episode to 10 episodes significantly improves Trojan detection performance across all feature combinations by 4% to 8%. This suggests that a Trojan detection requires many observations to overcome noise and could involve long-tailed key states that are not encountered in few episodes.

For feature combination, we observe that attribution of value networks $\nabla V(X)$ does not significantly improve performance when combined with $\nabla \pi(X)$. Instead, using scalar value function output has significant improvement from 0.9627 to 0.9728 AUC which stemmed from our observation that the value function carries distinct signature as shown in Figure 3. Finally, entropy $\mathbb{H}[\pi(X)]$ provides orthogonal improvements on top of attribution signatures, which we leave to future works to explore.

# 6   Conclusion

We demonstrate the challenging nature of poisoning attacks in RL agents, highlighting the limited existing research on defenses against such attacks. In this paper, we propose a practical attribution-based model analysis approach for the detection of Trojaned models. Our approach is characterized by its simplicity and efficiency, while also exhibiting generalization across different environments, model architectures, and trigger patterns. Furthermore, our approach achieves top performance across two IARPA RL datasets. As part of our future work, we plan to investigate the effectiveness of our method in multi-agent environments and more complex RL scenarios.

## Acknowledgments

The authors acknowledge support from IARPA TrojAI under contract W911NF-20-C-0038. The views, opinions and/or findings expressed are those of the author(s) and should not be construed as representing the official views or policies of the Department of Defense or the U.S. Government.

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
