# OpenReview forum: "Universal Trojan Signatures in Reinforcement Learning"
_NeurIPS.cc/2023/Workshop/BUGS — NeurIPS 2023 BUGS Poster_

### Official Review · Reviewer_9aoU · 2023-10-26
**Interesting work but needs further validation**

**Rating:** 6
**Confidence:** 4

**Review:**

This paper proposes a method for detecting trojans in reinforcement learning models. It leverages the observation that trojaned models exhibit distinct behavior even when given clean inputs. This is achieved by calculating the gradients of both the policy network and the value network with respect to the input. These gradients are then used as features to train a binary classifier that determines whether a model is trojaned. The experiments were conducted using the TrojAI challenge datasets. The results demonstrate the effectiveness of the proposed detection approach, surpassing a baseline weight analysis method.

Strengths

1. This addresses a timely issue in detecting trojaned reinforcement learning models.
2. The example illustrated in Figure 2 is both intuitive and interesting.
3. The proposed detection method is reasonable and exhibits good performance.

Weaknesses

1. The proposed approach requires training on hundreds of models. It would be beneficial to observe how the detection performance varies with different numbers of training models.

2. There might be an adaptive attack aiming to match the attribution between the trojaned model and a clean model on clean inputs. The performance of the proposed detection approach under such circumstances is not entirely clear.

---

### Official Review · Reviewer_1JEz · 2023-10-27
**An interesting backdoor detection method for RL models**

**Rating:** 8
**Confidence:** 4

**Review:**

This paper proposes a new defense method for auditing backdoor in policy of RL agents. In general, backdoor attacks on deep RL is a topic of great practical importance, which is still very much underexplored.

The main idea underlying the proposed defense method is that compared to clean models, small input perturbations lead to signification changes in the advantage prediction of attacked models, as the backdoors are added to direct the advantage toward undesirable actions. This observation is reasonable and leads to simple backdoor detection methods based on Jacobians of policy and value networks.

Experiments show promising initial results on the backdoor detection performance, and interestingly, on generalization to new environments beyond the training. The later point is very relevant to applications, and future work should investigate more on sensitivity and generalization of proposed detection method when being transferred to new environments.

Overall, this paper considers an important and underexplored topic in backdoor attack, the proposed method is simple and well-motivated. I think this is an interesting contribution to the workshop.

---

### Decision · Program_Chairs · 2023-10-28

**Decision:**

Accept (Poster)

**Comment:**

The reviewers agree that the paper addresses an important security risk: backdoor attacks on deep RL. However, there are some concerns about access to the training datasets of models and adaptive attacks, and we hope that the authors can find these comments useful for future revisions.